



# Dynamics of He++ ions at interplanetary shocks

Olga V. Sapunova[1], Natalia L. Borodkova[1], Georgii N. Zastenker[1], Yuri I. Yermolaev[1]

[1] Space Research Institute of the Russian Academy of Sciences, Moscow, Russia

*Correspondence to*: Olga Sapunova (sapunova_olga@mail.ru)

**Abstract.** Variations of parameters of twice-ionized helium ions — $He^{++}$ ions or α-particles — in the solar wind plasma during the interplanetary shock front passage are investigated. We used the data measured by the BMSW (Bright Monitor of Solar Wind) instrument installed on the SPEKTR-R satellite, which operated since August 2011 to 2019 and registered 57 interplanetary shocks. According to received data, the parameters of $He^{++}$ ions were calculated: bulk velocity $V\alpha$, temperature $T\alpha$, absolute density $N\alpha$ and relative density (helium abundance) $N\alpha/Np$. The correlation of changes in helium abundance $N\alpha/Np$ with the parameters $\beta_i$, $\theta_{Bn}$ and $M_{MS}$ were investigated.

## 1. Introduction

Interplanetary shocks (IP) generated by solar flares and coronal mass ejections and propagated in the solar wind are one of the main agents transferring perturbations from the Sun to the Earth (e.g., Borrini et al., 1982; Volkmer and Neubauer, 1985; Borodkova, 1986). All parameters of the solar wind plasma — velocity, temperature, density, and magnetic field — change dramatically when the IP shock front passes. It is known that the two main components of the ionic composition of the solar wind are protons and twice-ionized helium ions ($He^{++}$ or α-particles). At the same time, the share of single-ionized helium ions in the solar wind does not exceed thousandths of the density of $He^{++}$ ions and is beyond the scope of this paper.

Before proceeding with our main results, we should give a brief description of some basic elements of the problem. Variations in the proton and $He^{++}$ ion parameters and the $N\alpha/Np$ are directly related to the properties of the Sun upper corona and the mechanisms of solar wind formation in it. Therefore, it is an important question to determine the relative density of helium relative to the main (proton) component (e.g., Ogilvie and Wilkerson, 1969; Formisano et al., 1970; Borovsky, 2008; Kasper et al., 2012; Yermolaev et al., 2020 and references therein).

Thus, the study of IP shocks and the time profiles of various solar wind components during their propagation become interrelated tasks aimed at solving a single problem. The study of the changes of $He^{++}$ ions on the IP shock fronts and its interaction with it was started in (Gosling et al., 1978), after which this issue was investigated both by modeling and by experimental data (e.g., Scholer and Terasawa, 1990; Scholer, 1990; Trattner and Scholer, 1991). Main part of experimental results have been obtained on the basis of magnetic field and proton measurements. However, it needs higher time resolution to investigate $He^{++}$ ion parameter fine structure in the immediate vicinity of the ramp.

It became possible with launch of the BMSW instrument as part of the PLASMA-F experiment onboard the SPEKTR-R
satellite. This instrument allowed to study the fine structure of the IP shock front with high time resolution of the instrument
— 0.031 s for the magnitude and direction of the solar wind ion flow and 1 s for the velocity, temperature and ion density
(for protons and for He$^{++}$ ions) (e.g., Nemecek, 2013; Zastenker et al., 2013; Safrankova et al., 2013b; Eselevich, 2017). The
aim of this brief article is to study variations of the density of the He$^{++}$ ions at the front of an interplanetary shock and to
detect changes in the He$^{++}$ ions parameters directly next to the ramp with a precision high time resolution.

## 2. Experimental data

For the study, we used data obtained by the BMSW plasma spectrometer, which operated since August 2011 to 2019.
BMSW is intended for measurements of the energy spectrum of ions in the range of 0.2–2.8 keV/Q, bulk velocity in the
range from 200 to 750 km/s, ion isotropic temperature from 1 to 100 eV and ion density from 1 to 100 cm$^{-3}$.

The time resolution of the BMSW instrument was 0.031 s for the ion flux magnitude and direction and 1 s for the
velocity, temperature, and density of protons and He$^{++}$ ions. Also, in some cases, it a mode was available for measuring the
velocity, temperature, and density of protons with a resolution of 0.031 s. A detailed description is given in (e.g., Nemecek,
2013; Zastenker et al., 2013; Safrankova et al., 2013b; Eselevich, 2017). During the operating period, the BMSW instrument
registered 57 IP shocks.

Time profiles of the main solar wind plasma parameters was investigated and supplemented with measurements of the
magnetic field by the MFI magnetometer (with time resolution of 0.092 s) on the WIND satellite, which is located in the
solar wind near the first libration point. In some cases, the data from the THEMIS-B/THEMIS-C or Cluster 1-4 satellites
were taken. The basic parameters of the magnetosonic Mach number $M_{MS}$, the angle $\theta_{Bn}$ between the magnetic field vector
and the front normal direction, the plasma beta $\beta_i$ — the ratio of gas pressure to the magnetic pressure — were calculated for
all IP shocks.

We also used data obtained using the 3DP instrument on the WIND satellite with a time resolution of 3 s (when it was
reliable) to compare time profiles of protons and He$^{++}$ ions of the solar wind at different points in space.

## 3. Examples of He$^{++}$ ion measurement by the BMSW instrument

An example of the IP shock with a clearly visible flux of He$^{++}$ ions is shown in Fig. 1. It presents the energy time
spectrogram of the solar wind ion flux registered by the BMSW instrument. Two populations are clearly distinguished on the
spectrogram and are indicated by arrows. Both populations, protons and He$^{++}$ ions, exist before, during, and after the shock
front passes. According to the BMSW instrument, it is possible to distinguish the distribution of ions if the temperature and
velocity of protons are small (Safrankova et al., 2013a).




The curve of the sensor current dependence on the voltage gives two separate peaks at low ion temperatures — the proton peak (approximately at ~500 V) and the $He^{++}$ ion peak (approximately at ~1000 V) — as shown in Fig. 2 by solid red

and blue lines accordingly. The solar wind plasma accelerates and heats up behind the IP shock front, as a result of which the proton peak and the $He^{++}$ ion peak expand and shift to the right on the sensor current–voltage curve. Since the measured energy range was limited by 3 keV, the second peak could be shifted beyond the measured energy range in case of high velocity.

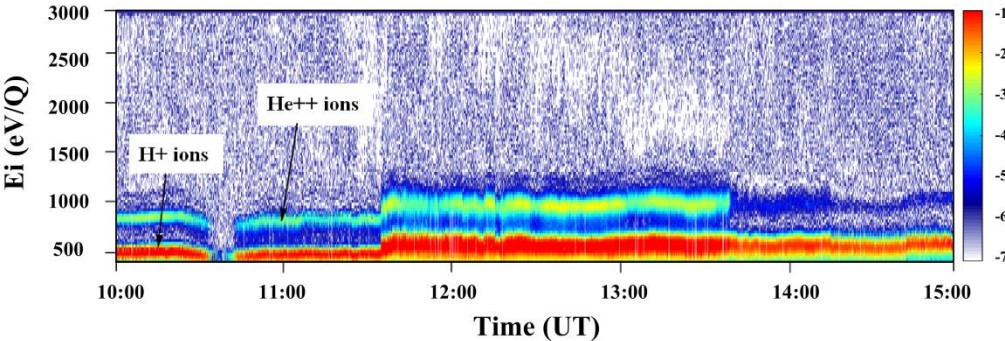

**Figure 1: Energy time spectrogram of ion flux measured on September 30, 2012 by the BMSW instrument.**

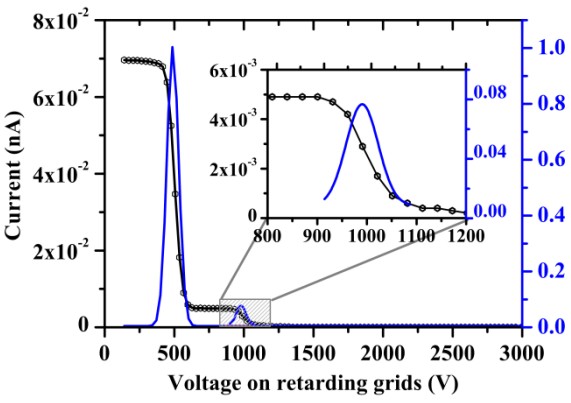

**Figure 2: An example of 3-second spectrum of the BMSW instrument. Black line – energy cutoff characteristic, blue line – derivative of FC currents for protons and $He^{++}$ ions.**

Thus, 20 interplanetary shocks were selected, for which it was possible to isolate the flow of $He^{++}$ ions during the passage

of the front. For each case, the parameters of protons and $He^{++}$ ions were calculated: bulk velocity ($V_p$, $V_\alpha$), temperature ($T_p$, $T_\alpha$), and density ($N_p$, for $He^{++}$ ions both $N_\alpha$ and helium abundance $N_\alpha/N_p$).





## 4. Comparison of values obtained from different satellites

In Fig. 3, an example of time profiles of main solar wind plasma parameters and the magnetic field during the IP shock passage on July 9, 2017 is shown. It represents the solar wind density, temperature, and proton velocity registered on the SPEKTR-R spacecraft (panel a-c) and the magnetic field magnitude registered on the WIND spacecraft (panel d-e) that was time-shifted to SPEKTR-R location. The profiles highlight the following structural features of the front: a ramp with a time size of ~0.34–0.35 s, the upstream oscillation wave train, with a wavelength having a time scale of ~0.37–0.47 s, and the downstream wave train with duration about ~0.64–0.84 s. The example illustrates a good agreement of the values obtained on two space satellites for the same IP shock: the duration of the IP shock ramp and the upstream and downstream wave train duration, determined by the parameters of the solar wind plasma and the magnetic field.

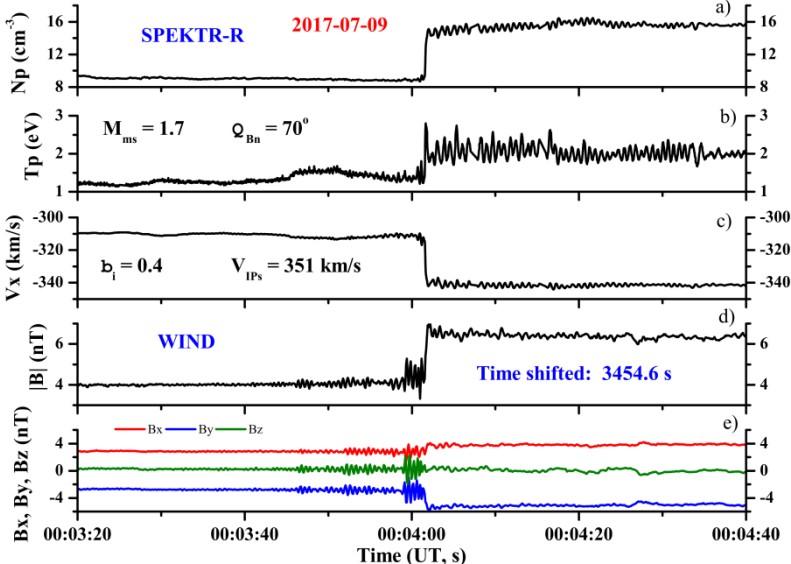

**Figure 3: Time profiles of plasma parameters and magnetic field during the IP shock passage on July 9, 2017. From top to bottom: density (a), temperature (b), and velocity (c) (along the Earth–Sun direction) of ion flow, magnetic field magnitude (d) and components (e) of solar wind plasma.**

Figure 4 represents an example of comparing the density (absolute — both for protons and $He^{++}$ ions, relative — for $He^{++}$ ions) obtained from the SPEKTR-R and WIND satellites for the very first IP shock in our database — September 9, 2011. There is a good general agreement of the measured parameters, in particular, a noticeable decrease of the $N\alpha/Np$ immediately after the IP front.



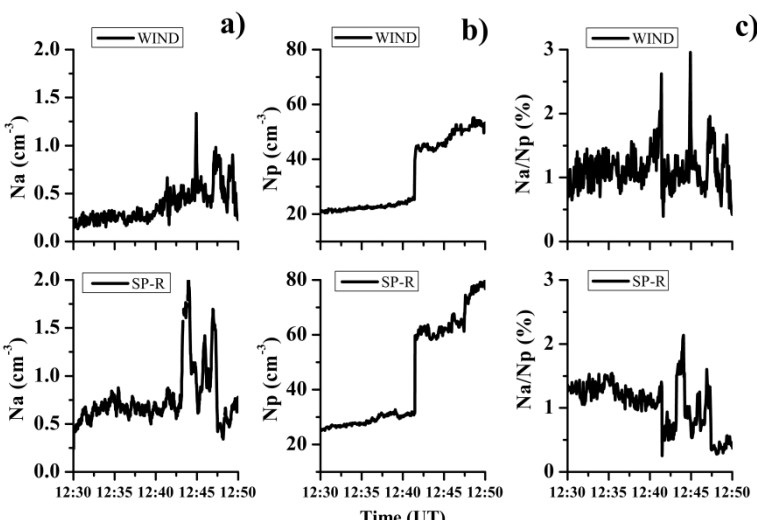

**Figure 4: Changes in time profiles of the proton and He++ ion density according to the SPEKTR-R and WIND satellites data during the event of September 9, 2011: absolute He++ ion density (column a); absolute proton density (column b); relative He++ ion density (column c).**

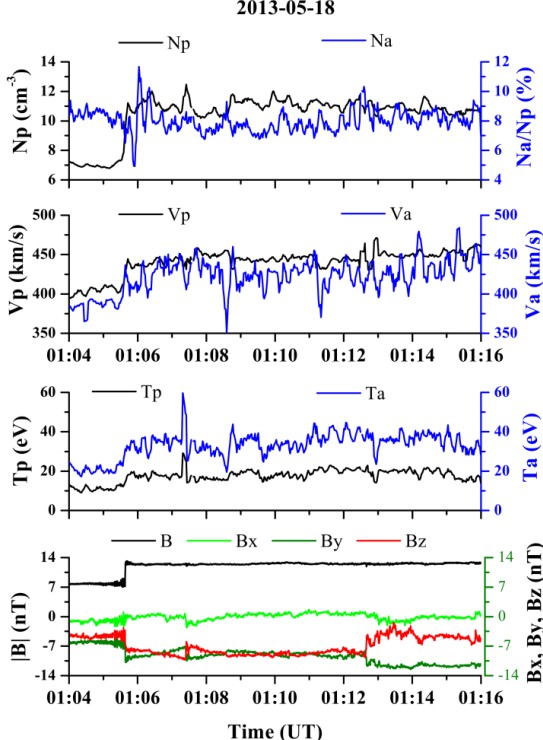

**Figure 5: Time profiles of solar wind plasma parameters and magnetic field during the IP shock crossing on May 18, 2013.**

## 5. Variations of the He$^{++}$ ion parameters during the IP shock passage

When the IP shock front passes, all parameters of solar wind change dramatically and significantly. An example of that changing is shown in Fig. 5 for density, velocity and temperature, as well as the magnetic field. One can see a slight decrease in the average Nα/Np immediately after crossing the IP shock front. The velocity of He$^{++}$ ions was, on average, slightly (about 7%) less than the velocity of protons both before and after the front, and the temperature of He$^{++}$ ions is 2 times higher than the temperature of protons.

Figure 6 shows the distribution of absolute Nα and Nα/Np for all events, before and after the front passage. It can be noted that after crossing the IP shock front, the absolute values change quite noticeably. In particular, the average value of the He$^{++}$ ion density increases by about 2 times. However, due to an equally strong increase in the density of protons behind the IP shock front, the Nα/Np changes slightly.

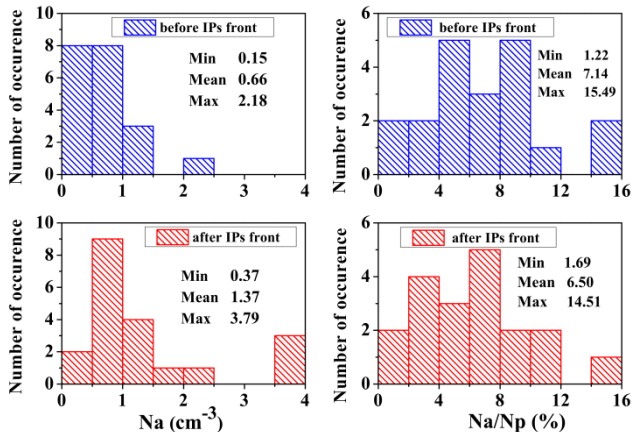

**Figure 6: Statistical histogram of the absolute (left) and relative (right) He++ ion density in undisturbed and disturbed solar wind.**

Because of the small number of events, we consider the change in the Nα/Np during the passage of the IP shock front depending on the values of $\beta_i$, $\theta_{Bn}$ and $M_{MS}$. The results are shown in Figure 7.

It can be noted that there is no explicit dependence of the change in the relative density of He$^{++}$ ions on the $\beta_i$ parameter (see Fig. 7b). For both small and large values of the parameter $\beta_i$, a large or small change in the relative density of He$^{++}$ ions is possible. There is also no clear correlation with the $M_{MS}$ parameter (see Fig. 7c). However, if we exclude an event with a large parameter value ($M_{MS} = 5.8$), there is a tendency for the relative density of He$^{++}$ ions to fall behind the IP shock front as the $M_{MS}$ increases. This issue requires further research.

However, we can see an obvious correlation between the change in the Nα/Np and the angle $\theta_{Bn}$ (see Fig. 7a). The smaller the angle $\theta_{Bn}$, the more the value falls. In other words, the Nα/Np will fall much more (2–2.5 times) behind the front of quasi-parallel IP shocks than behind the front of quasi-perpendicular IP shocks.



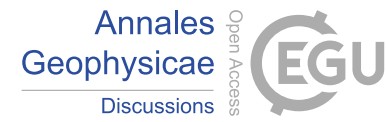

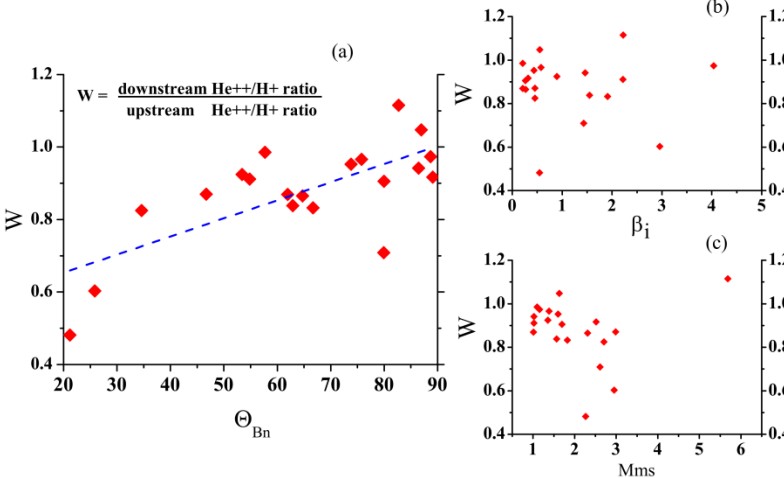

**Figure 7: Dependence of the He++ relative concentration change during the IP shock front passage on the $\theta_{Bn}$ (a), $\beta_i$ (b), and $M_{MS}$ (c) parameters.**

## 6. Discussion of results

The drop in the relative density of He$^{++}$ ions behind the IP shock front during the transition from a quasi-perpendicular to a quasi-parallel shock may be due to the outflow of some ions from the disturbed region to the undisturbed one through the ramp due to a decrease in the $\theta_{Bn}$ angle, which makes this transition more efficient. In the study (e.g., Trattner and Scholer, 1991), the time profile of reflected He$^{++}$ ions was modeled for quasi-parallel IP shocks. The diffusion of part of the ions (both protons and He$^{++}$) into the undisturbed region is noted, with the relative density of He$^{++}$ ions in a reflected stream that can be compared to that in an undisturbed solar wind. Also, it was shown for quasi-perpendicular cases too (e.g., Broll et al., 2017). This result is consistent with the one observed in the Fig. 7a dependence — in the case of a quasi-parallel IP shock, a significant part of the He$^{++}$ ions can move to an undisturbed region, which causes the relative density of He$^{++}$ ions behind the IP shock front to fall. The results published by Gosling et al. (1978) suggest that the mechanism of acceleration of low-energy ions on quasi-parallel IP shocks may be one-sided, which may also explain the result. For a more detailed study of this issue, it is necessary to increase the statistics of observation of quasi-parallel IP shocks.

In the case of a quasi-perpendicular IP shocks, the N$\alpha$/Np increases behind the IP shock front. It can be explained both by the difficulty of ion diffusion across the magnetic field and by changes in the mechanisms of nonlinear twisting of the IP shock ramp.

The problem of changes of the N$\alpha$/Np value in the case of IP shock with a large number of magnetosonic Mach remains open due to the insufficient number of events suitable for consideration.



## 7. Conclusion

All main parameters of $He^{++}$ ions (velocity $V\alpha$, temperature $T\alpha$, absolute $N\alpha$ and helium abundance $N\alpha/Np$) for 20
interplanetary shocks were calculated using the data of the BMSW instrument. It was shown that the average value of $N\alpha/Np$
behind the interplanetary shock front is slightly less (~9%) than in the undisturbed region, while the maximum value of this
parameter was even less behind the IP shock front. It should be noted again that this does not apply to the absolute value of
the $He^{++}$ ion density.

For a more detailed consideration of the time profiles of the $N\alpha/Np$, graphs of the dependence of this value on the
145 parameters $\beta_i$, $M_{MS}$, and angle $\theta_{Bn}$ were drawn. There was no obvious dependence of the change in the $N\alpha/Np$ on the
parameters $\beta_i$ and $M_{MS}$. It was revealed that a correlation exists between $N\alpha/Np$ and the angle $\theta_{Bn}$: the lower the value of the
angle $\theta_{Bn}$, the more the $N\alpha/Np$ falls behind the IP shock front.

The presented preliminary results were obtained with a small number of events and require further research.

### Acknowledgements

The authors express their gratitude to NASA CDAWEB for the possibility of using data on plasma and magnetic field
parameters measured on WIND and Cluster satellites. The work was supported by the Russian Science Foundation grant no.
16-12-10062.

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
