# Peer review of "Dynamics of He++ ions at interplanetary shocks"

_Annales Geophysicae, 2020_

## Referee Comment (RC1) · Anonymous Referee #1 · 26 Jan 2021

The authors of the article "Dynamics of He++ ions at the interplanetary shocks" use data from the BMSW instrument installed on the SPEKTR-R satellite to correlate the parameters of H++ with the shock parameters. The authors state in their article that "The present preliminary results were obtained with a small number of events and require further research". I fully agree with this statement and my recommendation is to complete their analysis and use a numerical code to understand better the physical process underway in the shock vicinity.

The article is not yet suitable for publication in its present form.
* * *

---

## Referee Comment (RC2) · Anonymous Referee #2 · 5 Feb 2021

General remarks:

In this paper the authors discuss preliminary results obtained by the BMSW instrument. Specifically, they focus on the change in the alpha-particle distribution during a limited number of shock crossings. Although the topic is relevant and of interest, the manuscript, in my mind, presents rather a "work in progress" and cannot be published in its present form.

A detailed list of comments, motivating this decision, it attached below.

More serious concerns:

1. As I understand, the BMSW instrument does not have a magnetometer on board and hence magnetic field measurements from other instruments on spatially separated spacecraft are used. These measurements from additional instruments, however,

make up a considerable number of the quantities considered including the Mach number, shock normal angle, etc. It is well-known that the turbulent magnetic field evolves over short timescales, and that the magnetic field (and especially fluctuations thereof) varies similarly over small length scales. This can also be seen from Figs. 4 and 5, where the plasma properties are significantly different between different instruments. Thus, I do not think that using magnetic field measurements from a different spacecraft can be used to analyse particles distributions measured on a another spacecraft. Therefore, in my mind, the bulk of the analysis presented in this manuscript cannot be robustly defended. If we then take away the magnetic field measurements, the manuscript contains insufficient new material for publication.

2. Only a small number of events are considered (a total of 20) and I think this small subset is insufficient for a statistical study. E.g. the results presented in Fig. 6 seem not to be statistically significant.

3. Given the small number of events (point 2), and the fact that non-local magnetic field measurements are used (point 1), I do not think that the results presented in Fig. 7 (and this is the main results of the manuscript) is statistically significant, and therefore not a robust result.

Smaller issues:

1. The manuscript states that IPs are generated by solar flares, which is definitely not correct.

2. Although I'm not a native speaker, the language in the manuscript has to be improved, and at several places, I had a hard time figuring out what the authors mean, e.g. "with a wavelength having a time scale of . . ."

---

## Referee Comment (RC3) · Anonymous Referee #3 · 2 Mar 2021

This paper presents preliminary results using He++ ion high time resolution measurements of the solar wind during the interplanetary shock front passage combined with magnetic field data. Either more information about the study performed should be included if this is to be an individual event study or else more events are needed if it is intended to be a statistical study. The English needs to be read/ edited by somebody in the field. This may help to clarify some issues in the text. Below some suggestions for the authors to consider.

Abstract: It could be added that BMSW data has high time resolution and that He++ measurements were compared with magnetic field data, and that 20 out of the 57 registered events were selected. At the end of the abstract 1-2 sentences presenting the main results would benefit the reader.

Page 1: The background and the motivation behind the work performed should be

elaborated in the introduction. At the end of the introduction shortly present the next sections of the paper.

Page 1: IT IS WRITTEN "Interplanetary shocks (IP) generated by solar flares and coronal mass ejections and propagated in the solar wind are one of the main agents transferring perturbations from the Sun to the Earth (e.g., Borrini et al., 1982; Volkmer and Neubauer, 1985; Borodkova, 1986)." The relevance of referring to both solar flares and coronal mass ejections as the origin of IPs needs to be explained in the context of the paper. (see for example Gopalswamy et al. 1998, https://agupubs.onlinelibrary.wiley.com/doi/epdf/10.1029/97JA02634).

PAGE 1: It would be useful if the references "(e.g., Scholer and Terasawa, 1990; Scholer, 1990; Trattner and Scholer, 1991)." could be separated between "....modeling (REF1, REF2) and by experimental data (REF 1, REF2). Which studies have compared outputs from both approaches?

Page 2: IT IS WRITTEN "The aim of this brief article is to study variations of the density of the He++ ions at the front of an interplanetary shock and to detect changes in the He++ ions parameters directly next to the ramp with a precision high time resolution." Why is this important? What does it teach us? Does it provide us information about the origin of the IP event?

Page 3: IT IS WRITTEN "Thus, 20 interplanetary shocks were selected, for which it was possible to isolate the flow of He++ ions during the passage of the front." Could the origin of the IP events have anything to do with this selection? 20 out of 57 IP registered events is not a high ratio. Need to better explain why it was not always possible to isolate the flow of He++ ions during the passage of the front. A table presenting and comparing the characteristics of the 20 IP events could be useful.

Page 8: IT IS WRITTEN "The presented preliminary results were obtained with a small number of events and require further research." For example (data analysis, modelling)? Though the current study used high time resolution measurements it is not

completely clear for me what the investigation has provided (the novelty of the results). This should be presented in the conclusion.

---

## Author Comment (AC1) · 24 Mar 2021

**Authors' response to reviewer#3 comments on manuscript**
**Dynamics of He++ ions at interplanetary shocks**
**by Sapunova et al.**

Authors thanks a lot to the reviewer for the review. We tried to improve our manuscript despite the lack of specific comments. We increased the number of cases by adding the Earth's bow shock crossings, so our results have become more statistically significant. For more details, you can also see the answer to reviewer#2. It contains extended information about changes which we brought into the article.

Best regards,
Olga Sapunova

---

## Author Comment (AC2) · 24 Mar 2021

**Authors' response to reviewer#3 comments on manuscript**
**Dynamics of He++ ions at interplanetary shocks**
**by Sapunova et al.**

**Dear Reviewer, Thank you very much for your remark comments in the interactive discussion of our paper. We tried to clarify points raised in your review and apologize if we didn't understand completely some of them.**

**Remark 1.** As I understand, the BMSW instrument does not have a magnetometer onboard and hence magnetic field measurements from other instruments on spatially separated spacecraft are used. These measurements from additional instruments, however, make up a considerable number of the quantities considered including the Mach number, shock normal angle, etc. It is well-known that the turbulent magnetic field evolves over short timescales, and that the magnetic field (and especially fluctuations thereof) varies similarly over small length scales. This can also be seen from Figs. 4 and 5, where the plasma properties are significantly different between different instruments. Thus, I do not think that using magnetic field measurements from a different spacecraft can be used to analyse particles distributions measured on a another spacecraft. Therefore, in my mind, the bulk of the analysis presented in this manuscript cannot be robustly defended. If we then take away the magnetic field measurements, the manuscript contains insufficient new material for publication.

**Reply to Remark 1. (This text will be also added to the)**

Magnetometer was installed onboard the SPEKTR-R satellite, but, unfortunately, didn't operate. We agree that magnetic field and plasma fluctuations occur in the solar wind and shock front structures observed on the WIND and SPEKTR-R may not be identical. However, as it was shown in (*Weygand, J.M., Matthaeus, W. H., Kivelson, M. G., Dasso S., 2013, Magnetic correlation functions in the slow and fast solar wind in the Eulerian reference frame. Journal of Geophysical Research: Space Physics, 118, 3995–4004. doi:10.1002/jgra.50398*) that the assumption that the IP magnetic field fluctuations are frozen in at distances from the L1 Lagrange point to the Earth is valid, while over large distances, the frozen-in assumption will break down first for the fast solar wind. So the presence of magnetic field fluctuations in the slow solar wind has little effect on its quasi-stationary structure. At the same time, it is known that IP shocks usually propagate in the slow solar wind which has fluctuations in density. Matthaeus et al. *(Matthaeus et al., 2016)* investigated space-time correlation of plasma turbulence in the solar wind and revealed that the plasma frame slow wind correlation persists for larger time separation. Also, there are inhomogeneities along the front itself. In such a case, the collisionless shock front structures measured by SPEKTR-R and WIND might differ noticeably from each other. According to Eselevich and Eselevich, *(Eselevich, M. V., &Eselevich, V. G., 2005, Fractal Structure of the Heliospheric Plasma Sheet in the Earth's Orbit. Geomagnetism and Aeronomy, 45, 3, 326–336.)*, the spatial scale at which the solar wind can be considered as uniform in density along the shock front in the ecliptic plane, is about $(4 -8) \cdot 10^6$ km. This is a fairly large size along the shock front. Taking into account that the properties of He++ ions are investigated on the MHD scales magnetic field measurements onboard WIND can be used for shock analysis.

**We took into account your comment about Figures 4 and 5 and expanded the description of Figure 4 (in new version this figure will change the number to 2 due to the revision of the article structure) to present our position:**

Figure 4 represents an example of comparing the densities (absolute — both for protons and He++ ions, relative — for He++ ions) obtained on the SPEKTR-R and WIND satellites for the very first IP shock in our database — September 9, 2011.

Data of both instruments show structures according to the parameters of the density of He++ ions. In column a), the blue arrows indicate 3 structures with an increased absolute density of He++ ions. They are located downstream, although their position relative to the IP shock ramp is slightly different, the coincidence of the shapes and the number of peaks indicates a good stability of the structures, given that we are talking about a perturbed region. A certain difference in the absolute values of the proton density, which is noticeable in column b), can be explained by the different sensitivity of the sensors of the instruments. Despite this, it should be noted that the relative change in the proton density coincides on both instruments and amounts $Np2/Np1 = 1.85$. The helium abundance $N\alpha/Np$ is given in column c), representing three structures corresponding to those shown in column a). Also, the red arrows highlight the increase in the helium abundance $N\alpha/Np$ immediately before the IP shock ramp, which is clearly visible from the data of the two instruments, as well as a sharp decline after ramp.

[Figure]

**We changed Figure 5 by adding the proton density, velocity and temperature of the solar wind measured by the 3DP instrument installed on the WIND spacecraft for comparison.**

Figure 5 shows that, despite the small fluctuations in the values, the velocity of the solar wind protons measured at different points in space coincides in numerical values. As for the protons density measured by different spacecraft, it should be taken into account the fact that instruments with sensors based on the Faraday cup measure the density much more accurately than electrostatic analyzers with detectors based on MCP. And for the proton temperature, the situation is opposite.

Thus, a comparison of the parameters of the plasma and the magnetic field measured at different points in space shows that there are both well-matched areas and different ones. Taking into account the fact that the dynamics of the behavior of He++ ions is studied on the MHD scale, the authors believe that it is possible to combine magnetic field measurements on WIND spacecraft with plasma measurements on SPEKTR-R satellite.

[Figure]

**We expanded the description of Figure 5 and the following paragraph was inserted into the text of the article:**

Figure 5 (panels a-c) shows the plasma data of the SPEKTR-P and WIND satellites measurements. The proton parameters are given in black for BMSW and in orange for the 3DP instruments, respectively. The blue color shows the parameters of alpha-particles according to the data of the BMSW instrument. Panel d of Fig. 5 shows the magnitude and components of the magnetic field according to the MFI instrument of the WIND satellite. There are areas with a good coincidence of parameters, and also areas with some differences in values, but in general, the data of the two instruments are similar in shape. It should be noted that the proton bulk velocity, measured by WIND and SPEKTR-R, are in good agreement with each other, and the slope of the IP ramp are the same for all proton parameters measured by different instruments. This fact confirms IP shock ramp stability on MHD scales during the shock propagation from WIND to SPEKTR-R.

**Remark 2.** Only a small number of events are considered (a total of 20) and I think this small subset is insufficient for a statistical study. E.g. the results presented in Fig. 6 seem not to be statistically significant.

**Remark 3.** Given the small number of events (point 2), and the fact that non-local magnetic field measurements are used (point 1), I do not think that the results presented in Fig.7 (and this is the main results of the manuscript) is statistically significant, and therefore not a robust result

**Reply to Remark 2 and 3. (This text will be also added to the article)** Unfortunately, due to the short period of the SPEKTR-R satellite, we are limited by the available amount of data recorded by the BMSW instrument.

We have increased statistics of events, for which we have processed data of the Earth's bow shock crossings by the SPEKTR-R in the period from 2011 to 2013.

To increase the statistics, we processed data of the Earth's bow shock crossings by the SPEKTR-R during the period from 2011 to 2013 year. The Figure 8 shows obtained results, superimposed with the existing data set. The estimation of the $\theta_{Bn}$ angle for the new events was made using the model (Verigin et al., 2003a.) for the shape of the bow shock and data from nearby satellites, including an estimate of the magnetic field direction. The new data set allowed us to supplement the area of quasi-parallel events, with an angle of $\theta_{Bn} < 45^{o}$. Despite the errors of this definition (shown in the Figure 8), there is a trend - the larger the $\theta_{Bn}$ angle is, the more helium abundance $N\alpha/Np$ will increase after IP shock front. This trend coincides with the one already mentioned in the set of the IP shock fronts crossings - the helium abundance $N\alpha/Np$ changes less in the quasi-parallel cases.

However, comparisons of the two sets of events show a significant difference in the values of the helium abundance $N\alpha/Np$ change. At the IP shock front crossings, the helium abundance $N\alpha/Np$ usually becomes less than in the unperturbed region, but in the case of the Earth's bow shock crossing, this parameter always increases, in some cases - by almost an order of magnitude. The literature usually describes the results of simulations performed for ions reflected from the front, but not for those that have passed beyond the ramp. However, a recent paper (Ofman et al., 2019) showed the possibility of a strong increase in the helium abundance $N\alpha/Np$ behind the shock front. The data obtained by us are consistent with the results of modeling performed in this work.

[Figure]

Figure 8 The dependence of the helium abundance Nα/Np change on the $\theta_{Bn}$ angle at the intersection of the shock front. Red marks show the events of the IP shock fronts crossings, black marks - the Earth's bow shock front crossings. The blue dashed line shows the trend for the first set, and the purple dashed line shows the trend for the second set. Errors in determining the $\theta_{Bn}$ angle are shown by bars.

**Smaller issues:**

1. The manuscript states that IPs are generated by solar flares, which is definitely not correct.

The interplanetary shocks are generated by fast phenomena of SW plasma (usually by two types pushing like a piston -  High Speed Streams or fast ICMEs) when  velocity difference between piston and undisturbed solar wind is higher than sound or Alfvenic speeds (e.g. Dryer, 1994; Berdichevsky et al., 2000; and references therein).

**We are thankful to the reviewer for his/her indication of the inaccuracy, which was made in the text. We updated this description in the introduction.**

2.  Although I'm not a native speaker, the language in the manuscript has to be improved, and at several places, I had a hard time figuring out what the authors mean, e.g. "with a wavelength having a time scale of..."

**We are grateful to the reviewer for his/her careful reading and comments about incorrect expressions and typos made during the layout of the article. We tried to fix all the errors we noticed.**

Best regards,
Olga Sapunova

---

## Author Comment (AC3) · 24 Mar 2021

**Authors' response to reviewer#3 comments on manuscript**
**Dynamics of He++ ions at interplanetary shocks**
**by Sapunova et al.**

**The authors are grateful to Reviewer for careful consideration of the manuscript and useful comments. Our reply is marked with initials OS and bold text.**

**Reviewer:** This paper presents preliminary results using He++ ion high time resolution measurements of the solar wind during the interplanetary shock front passage combined with magnetic field data. Either more information about the study performed should be included if this is to be an individual event study or else more events are needed if it is intended to be a statistical study. The English needs to be read/edited by somebody in the field. This may help to clarify some issues in the text. Below some suggestions for the authors to consider.

**OS: We took into account the comments of the reviewer, revised the manuscript and improved the English.**

**Reviewer:**
Abstract: It could be added that BMSW data has high time resolution and that He++ measurements were compared with magnetic field data... At the end of the abstract 1-2 sentences presenting the main results would benefit the reader.
**OS: The abstract was extended and now it includes information about data added from other satellites and main result obtained in the study.**

**Reviewer:**
...and that 20 out of the 57 registered events were selected...
**OS: These details were added to the text in section 3.3: "So, high temperature and high velocity are two main reasons why not all 57 IP shocks were suitable for $He^{++}$ parameters definition. Thus, 20 interplanetary shocks ... were selected..."**

**Reviewer:**
Page 1: The background and the motivation behind the work performed should be elaborated in the introduction. At the end of the introduction shortly present the next sections of the paper.
**OS: We expanded the introduction and methodology sections. We added links and explanations; also at the end of the introduction we shortly present the structure of the paper.**

**Reviewer:**
Page 1: IT IS WRITTEN "Interplanetary shocks (IP) generated by solar flares and coronal mass ejections and propagated in the solar wind are one of the main agents transferring perturbations from the Sun to the Earth (e.g., Borrini et al., 1982; Volkmer and Neubauer, 1985; Borodkova, 1986)." The relevance of referring to both solar flares and coronal mass ejections as the origin of IPs needs to be explained in the context of the paper. (see for example Gopalswamy et al. 1998, https://agupubs.onlinelibrary.wiley.com/doi/epdf/10.1029/97JA02634).
**OS: We corrected inaccuracy made in the IP description and added references:**
**"Interplanetary shocks (IP) (including generated by so-called High-speed streams from coronal holes and coronal mass ejections) propagating in the solar wind are one of the main agents transferring perturbations from the Sun to the Earth (e.g., Borrini et al., 1982; Volkmer and Neubauer, 1985; Borodkova, 1986; Yue et al., 2010; Ma et al., 2019). "**

**Reviewer:**

PAGE 1: It would be useful if the references "(e.g., Scholer and Terasawa, 1990; Scholer, 1990; Trattner and Scholer, 1991)." could be separated between "....modeling (REF1, REF2) and by experimental data (REF 1, REF2). Which studies have compared outputs from both approaches?

**OS: We added references and separated them:**

**"The study of the changes of He$^{++}$ ions on the IP shock fronts and their interaction was started by Gosling et al., (1978), after which this issue was investigated both by modeling (e.g. Scholer and Terasawa, 1990; e.g. Scholer, 1990; Trattner and Scholer, 1991) and by experimental (e.g., Borrini et al., 1982; Volkmer and Neubauer, 1985; Borodkova, 1986)."**

**Reviewer:**

Page 2: IT IS WRITTEN "The aim of this brief article is to study variations of the density of the He++ ions at the front of an interplanetary shock and to detect changes in the He++ ions parameters directly next to the ramp with a precision high time resolution." Why is this important? What does it teach us? Does it provide us information about the origin of the IP event?

**OS: We included discussion of these problem in the manuscript:**

**"Variations in the proton and He$^{++}$ ion parameters and the Nα/Np at large-scale distances > 10$^6$ km are directly related to the properties of the Sun upper corona and the mechanisms of solar wind formation in it. Therefore, it is an important problem to determine the relative density of helium relative to the main (proton) component variations due to local physical processes at small-scale distance ~10$^3$ km (e.g., Ogilvie and Wilkerson, 1969; Formisano et al., 1970; Borovsky, 2008; Kasper et al., 2012; Safrankova et al., 2013a; Yermolaev et al., 2020 and references therein)."**

**Reviewer:**

Page 3: IT IS WRITTEN "Thus, 20 interplanetary shocks were selected, for which it was possible to isolate the flow of He++ ions during the passage of the front." Could the origin of the IP events have anything to do with this selection? 20 out of 57 IP registered events is not a high ratio. Need to better explain why it was not always possible to isolate the flow of He++ ions during the passage of the front. A table presenting and comparing the characteristics of the 20 IP events could be useful.

**OS: This information was added to the text in section 3.3: "So, high temperature and high velocity are two main reasons why not all 57 IP shocks were suitable for He$^{++}$ parameters definition. Thus, 20 interplanetary shocks ... were selected..."**

**Also the new set of cases was added to the research to improve reliability of the results and a following table of main parameters was included:**

**Table 1 Parameters of IP shock and Earth's bow shock crossing.**

| IP shock crossings | | | | | | Earth's bow shock crossings | | |
|---|---|---|---|---|---|---|---|---|
| Date | $V_{IP}$ | $\beta_p$ | $\theta_{Bn}$ | $M_{MS}$ | Na2(%)/ Na1(%) | Date | $\theta_{Bn}$ | Na2(%)/ Na1(%) |
| 09.09.2011 | 412 | 3.0 | 26 | 3.0 | 0.6 | 23.03.2012 | 77±4 | 7.3 |
| 01.11.2011 | 403 | 0.4 | 74 | 1.5 | 0.95 | 28.03.2012 | 39±4 | 1.8 |
| 15.05.2012 | 428 | 1.5 | 86 | 1.0 | 0.94 | 05.04.2012 | 71±5 | 3.5 |
| 21.05.2012 | 406 | 1.4 | 80 | 2.6 | 0.71 | 23.04.2012 | 52±2 | 4.5 |
| 03.09.2012 | 457 | 0.5 | 35 | 2.7 | 0.82 | 28.05.2012 | 21±2 | 3.1 |
| 30.09.2012 | 302 | 1.8 | 65 | 1.8 | 0.83 | 07.08.2012 | 81±5 | 5.1 |
| 08.10.2012 | 409 | 0.3 | 84 | 1.7 | 0.91 | 08.08.2012 | 65±5 | 9.5 |
| 13.04.2013 | 472 | 0.5 | 47 | 3.0 | 0.87 | 24.08.2012 | 85±5 | 11.0 |
| 23.04.2013 | 312 | 1.5 | 63 | 1.6 | 0.84 | 16.09.2012 | 45±4 | 4.1 |
| 18.05.2013 | 502 | 0.2 | 75 | 1.3 | 0.99 | 12.10.2012 | 88±4 | 7.2 |
| 19.04.2014 | 520 | 0.2 | 62 | 1.0 | 0.87 | 30.10.2012 | 76±5 | 4.4 |
| 03.05.2014 | 225 | 4.0 | 89 | 1.2 | 0.97 | 02.11.2012 | 29±5 | 2.0 |
| 07.06.2014 | 438 | 0.3 | 89 | 2.5 | 0.92 | 14.11.2012 | 84±4 | 5.6 |
| 03.07.2014 | 309 | 2.2 | 55 | 1.0 | 0.91 | 17.11.2012 | 64±3 | 7.3 |

| 17.03.2015 | 562 | 0.3 | 65 | 2.3 | 0.87 | 24.11.2012 | 83±2 | 5.0 |
|---|---|---|---|---|---|---|---|---|
| 21.06.2015 | 327 | 2.2 | 83 | 5.7 | 1.12 | 09.03.2013 | 84±5 | 6.9 |
| 12.10.2016 | 431 | 0.5 | 21 | 2.3 | 0.48 | 11.03.2013 | 80±4 | 4.2 |
| 09.11.2016 | 354 | 0.6 | 87 | 1.6 | 1.05 | 11.03.2013 | 60±4 | 2.6 |
| 31.08.2017 | 398 | 0.9 | 53 | 1.4 | 0.92 | 14.03.2013 | 76±4 | 7.9 |
| 21.10.2017 | 395 | 0.6 | 76 | 1.4 | 0.97 | 16.05.2013 | 22±5 | 5.3 |
| -- | -- | -- | -- | -- | -- | 09.06.2013 | 48±4 | 3.8 |
| -- | -- | -- | -- | -- | -- | 10.06.2013 | 27±4 | 3.5 |
| -- | -- | -- | -- | -- | -- | 06.07.2013 | 32±6 | 2.5 |
| -- | -- | -- | -- | -- | -- | 13.09.2013 | 50±4 | 7.7 |
| -- | -- | -- | -- | -- | -- | 01.12.2013 | 35±5 | 6.6 |

**Reviewer:**

Page 8: IT IS WRITTEN "The presented preliminary results were obtained with a small number of events and require further research." For example (data analysis, modelling)? Though the current study used high time resolution measurements it is not completely clear for me what the investigation has provided (the novelty of the results). This should be presented in the conclusion.

**OS: The Conclusion part was also updated and, in particular, the following part was added:**

**"It was revealed that a correlation exists between Nα/Np and the angle $\theta_{Bn}$: the lower the value of the angle $\theta_{Bn}$, the more the helium abundance Nα/Np falls behind the IP shock front. For Earth's bow shock crossings it was shown a significant increase of the helium abundance Nα/Np in quasi-perpendicular events. These results correspond with ones, showed by Ofman et al. (2019)."**

Best regards,
Olga Sapunova